# The battle to achieve Sustainable Development Goal Two: The role of environmental sustainability and government institutions

Justice Gyimah[1]*, Benjamin M. Saalidong[2], Louis K. M. Nibonmua[3]

1 College of Economics and Management, Taiyuan University of Technology, Taiyuan, China, 2 Yale School of the Environment, Yale University, New Haven, Connecticut, United States of America, 3 Department of Supply Chain and Information Systems, School of Business, Kwame Nkrumah University of Science and Technology, Kumasi, Ghana

* gyimahjustice@gmail.com

**Data Availability Statement:** The data for the study is from World Development Indicators (https://databank.worldbank.org/source/world-development-indicators).

## Abstract

The current period marked by addressing environmental sustainability challenges and the instability of government institutions has heightened the issue of food security, especially in developing countries as they work towards achieving Zero Hunger as highlighted in the Sustainable Development Goals. To assess the effect of environmental sustainability and government institutions on food security in West Africa with data from 1990 to 2021, two models have been deployed. The Generalized Method of Moments was deployed as the main model and while Two-Stage Least Squares was used as the robustness check. The findings of the study reveal that carbon emissions which represent environmental sustainability has no direct significant effect on food security, while government institutions has negative effect on food security. The study also reveals that income and urbanization promote food security, while renewable energy and population growth reduce food security. The findings of the study could be a reflection of the current political instability and attitude towards tackling carbon emissions mitigation in the region. Government institutions are encouraged to exercise authority without fear to implement policies that would encourage food security and restrict the use of high-emission technologies.

## Introduction

Estimates predict that by 2050, there will be an increase in the world population by two billion, which will bring the total world population to about 10 billion [1]. This increase in population growth is certainly going to bring a high demand on the limited world resources, most notably on food supplies. In the era of globalization coupled with the projected increase in world population, there need to attain global food security together with poverty reduction becomes a global challenge for humanity in achieving sustainability. This poses a threat to the second global sustainable development goal (no hunger), attaining food and its related environmental security comes with significant challenges. The Food and Agriculture Organization of the United Nations (FAO), together with the World Food Program (WFP) defines food security to

**Funding:** The authors received no specific funding for this work.

**Competing interests:** The authors have declared that no competing interests exist.

be a situation whereby "all people, at all times, have physical, social, and economic access to sufficient, safe, and nutritious food that meets their food preferences and dietary needs for an active and healthy life" [2].

Food security remains one of the top priorities of the United Nation's Sustainable Development Goals (SDG) which is aimed at alleviating hunger (zero Hunger) by 2030 [3, 4]. In 2018, the Food and Agriculture Organization (FAO) of the United Nations estimated that approximately 821 million people across the world were food insecure [5, 6]. The number of food-insecure people in 2022 is estimated at 1.3 billion, an increase of 10 percent from Economic Research Service's 2021 estimate [7]. Eliminating food insecurity and malnutrition in all forms by 2025 has been the main target of the African Union's Agenda 2063 [8]. Achieving this goal comes with its own challenges in light of the need to feed more people from the expected increasing global population coupled with the impacts of climatic changes on food production. Additionally, the current COVID-19 pandemic which comes with negative economic impacts, together with the recent large-scale desert locus outbreaks in many parts of Africa is expected to exacerbate the number of people undergoing malnutritional conditions. As at 2022, the number of people undernourished in Africa was 282 million, which is about 46 million more people compared with 2019 [9, 10]. This presents a heavy blow in achieving the zero-hunger goal by 2030 as set by the United Nation.

Access to sufficient food is one of the most basic and important human needs, however, hundreds of millions of people suffer from starvation, with over 25,000 succumbing to hunger daily [11]. Access to adequate food has been considered to be a basic human right after the 1943s Hot Springs Conference on Food and Agriculture [12, 13]. However, about one billion of the global population are currently undernourished [14] while over two billion people suffer from a lack of essential nutrients in their food with nearly six million children suffering from malnutritional related diseases every year [15]. In recent years, living standards of some sub-Saharan African countries have improved significantly [16]. This can be associated with growth in agricultural gross domestic product (GDP) [17], improved health [18] and better nutrition awareness or education [19]. Regardless of these positive trends in living standards, food insecurity still remains a major challenge facing these countries. The study by Frelat et al [20] reveals that over 30% of rural households across 17 countries within the sub–Saharan Africa are still struggling with food insecurity.

The exacerbation of the current challenges facing food security could also be attributed to the emergence of the COVID-19 pandemic coupled with the ongoing conflict, economic turbulence, and the consequences of climate change. While the cause of food insecurity and related hunger may vary from country to country, food insecurity is generally a result of economic shocks, environmental issues as well as conflicts, and humanitarian crises [21]. These contributing factors in combination with the growing global population which results in an increase in demand for resource-intensive and environmentally impactful food, are putting a strain on the planet's resources. The elimination of food insecurity, and poverty and the management of natural resources are major policy goals for many African countries in achieving their sustainable development goals. To be successful in eliminating food insecurity in the face of population growth and climate changes will require an increase in food production in sub–the Saharan Africa region. The scaling up of food security has met with different arena challenges arising from economic, political, engineering, and agronomics.

Achieving food security requires a concerted effort from both environmental sustainability practices and effective government institutions. Food security remains a critical issue in the face of global population growth and the threat of climate change. The collaboration between government institutions and private organizations has an impact on the development of effective food security strategies. Biodiversity preservation is essential for resilient food systems,

this help crops and livestock to be better equipped to withstand environmental challenges such as climate change. Climate change has an impact on agriculture, leading to more frequent extreme weather events, changing growing seasons, and affecting crop yields. There is the need to develop resilient agricultural practices to produce food sustainably, minimizing negative environmental impacts. In adopting sustainable agricultural practices and implementing supportive policies and programs, governmental institutions will have a vital role in the reliability of food supply while safeguarding the environment for future generations. This study aims at assessing the feedback direct effects of carbon emissions and governance institutions on food security and vice versa within the sub-Saharan sub-region. This study will help make informed decisions, design effective policies, and implement sustainable practices to ensure food availability, accessibility, and affordability for current and future generations. This study will explore the current state of food security within sub-Saharan Africa and the connection of food security with various contributing factors such as economic growth, governance, geopolitical, climatic, population, and environmental factors. This study will also contribute to existing knowledge on food security within the sub-Saharan region by revealing the state of food security and its controlling factors at the regional level. All these factors in combination with population and climatic conditions contribute to unstable food production and fluctuations in food prices.

The rest of the study has been structured, section 2 literature review, section 3 methodology, section 4 results and discussions, and section 5 conclusion and policy implication.

## Literature review

Changes in climate conditions and the scarcity of natural resource is a known factor that is making the ability to meet the increased demand for food even more challenging [2]. Climate change has the potential to affect crops and livestock productivity. Food security in sub-Saharan Africa is mostly attributed to climatic-related issues, most local farmers in the sub-Saharan region perceive crop failures to be caused by a lack of rainfall. The majority of sub-Saharan Africa's cultivated land is rain fed. The impacts of climate change such as an increase in temperature, changes in rainfall patterns, high atmospheric carbon dioxide, and more intense and frequent extreme weather events, could have a negative impact on agriculture. It has been estimated that, due to global warming, global agricultural productivity will decrease between 3 and 16% by the 2080s, this decrease could vary between 10 and 25% in developing countries [22]. Forest cover is also an important factor in managing climatic conditions through the provision of ecosystem services such as water quantity and quality management, soil erosion reduction as well as the provision of conducive micro-climatic conditions that promote productivity [23]. Forest cover also plays an important role in the control of climate change through carbon sequestration.

A study by Twongyirwe et al [24] in Uganda reveals that the majority of respondents attributed food insecurity in the country to drought. In as much as drought may be a common factor within the sub-Saharan region where food insecurity is experienced, marginal and drought-prone areas are known to be vulnerable to crop failure. Food insecurity is influenced by more than just drought. In the study Lewis [25] where the effect of drought on crop failure and food insecurity was isolated, the study found that the lack of resilience on the part of smallholder farmers and pastoralists as well as the lack of timely warning systems accounted for the majority of local food insecurity in Ethiopia. Similarly, the work of Amwata et al [26] reveals that, in marginal areas of Kenya, households with agropastoral livelihood are less vulnerable in terms of food security as compared to their pure pastoral counterparts. Turyahabwe et al [27] studies on how wetland resources contribute to household food security in agroecological zone in

Uganda. The study reveals a high rate of food insecurity (83%) to be associated with loss of wetland resource to farming, the study also found a high dependence rate (80% of households) on wetland resource. These studies suggest that, the rate of food insecurity in marginal areas can be influenced by off-farming employment activities and inadequate access to climate information. This also implies that there is a need for policies that will help promote access to climate information and diversification of livelihoods and adequate access to production resources. Food insecurity can also be viewed as inter-temporal in nature which is associated with socio-demographic and other topographical conditions. Living in certain topographical or ecological zones could play a key role in food security issues, as a result of limited opportunities or limitations which are naturally associated with such zones. This is confirmed by the studies of Bolarinwa et al. [28], in their study, it was observed that households experience a transitional period of insufficient food security during planting and growing seasons, which then evolves into a state of complete assurance of food availability during the harvesting periods.

Governance and institutional structure can also play an important role in food production. Bad governance can negatively affect the growth and productivity of the food sector, poor institutional structures such as corruption hugely affect farmer's productivity especially when it comes to the subsidization of agricultural materials such fertilizer, seed, agricultural pricing, and transportation [29, 30]. Bad governance can slow down agricultural production and even raise the cost of production as against the rise in taxes [31]. According to the study by Drebee & Abdul-Razak [32] quality governance is an important component in food production which accounts for about 29% of the variation in the farming sector. The level of effectiveness in agriculture sector reflects the degree of commitment governments have toward policies for improving agriculture sector. The security of food supply can be promoted through the effective formulation and regulation of government policies for production, distribution, and management [33]. Quality institutional governance can also help to promote the protection of the environment through the minimization of carbon dioxide emissions and consequently promote ecological quality [34].

The increase in population growth certainly has a connection to food security, ranging from the changes in human diets to the methods and ways through which food is produced [35]. For instance, the increase in population will also mean more people are becoming wealthier and thus will be able to eat more and regularly or more people turn to opt for more resource-intensive food, thereby leading to the decline of resources needed for food production. Meeting the ever-increasing demand for food as a result of the increase in population will mean that food production needs to be scaled up. Meanwhile, the current agricultural system is struggling to deliver enough food to meet the need of the current population in many parts of the world most especially in Africa [36]. Food insecurity challenges get intense in areas of Sub-Saharan Africa with rural settings where means of living are largely dependent on agriculture, this has been confirmed in many studies [37, 38]. The level of food insecurity among households can be attributed to unavailability or lack of access to arable land (emanating from increasing population growth and competition for different land uses). The decrease in arable land can be attributed to rapid industrialization and urbanization. In a study conducted by Bren d'Amour et al [39], an increase in urbanization could result in a 1.8 to 2.4% loss of global croplands. This limits the ability of households to increase the scale of production.

Economic condition especially GDP per capita is one of the main drivers of global variation in rates of food insecurity. Economist has predicted that the number of people experiencing moderate or severe food insecurity is expected to increase in low-income countries. Developed economies are generally more conscious of their state of food security than emerging economies [40], this certainly comes with benefits by helping individuals to get food and prevent

famine. To ensure sustainable food provision, there will be a need for both macro and micro economic policy measures [41]. Economic constraints can also serve as a barrier to investment in the agriculture sector to increase crop production. Moreover, an increase in the price of agricultural products such as fertilizer coupled with low prices of farm products can also result in an unfavourable cost-benefit ratio [42, 43]. This unfavourable cost-to-profit ratio is mostly high in remote areas where there is poor access to fertilizer and other materials for farming. Socioeconomic parameters were also found to be important factors that influence household food resilience capacity during periods of economic hardships and other stressors. Yikii et al [44] attributed the high prevalence of food insecurity in 520 households to poverty, unemployment, and a high illiteracy rate. In terms of socio-demographic factors, Negesse et al [45] identified a relationship between female-headed households and the occurrence of food insecurity, with female-headed households being more vulnerable to food insecurity than male-headed households in Ethiopia. These factors further suggest that drought alone does not contribute to the fluctuations in household food production and consequently its food security.

Large-scale land acquisition will also help in expanding available areas for agricultural activities [46]. which will in turn help achieve food security. Land scarcity coupled with the complex land tenure system has been one of the constraints on the agriculture sector in many African countries. In Ghana for instance, out of 1.9 million hectares of potentially arable land, only 31000 hectares (1.6%) have been developed for agriculture production [47]. The scarcity of land for agricultural production could be attributed to rapid urbanization, which is fast displacing urban and peri-urban agriculture activities. Most potential lands for agricultural production are been taken over by estate developers for residential and commercial purposes, which are perceived to be more economically attractive than agriculture [48]. The complex nature of land tenure systems in Africa, especially Ghana can also limit access to land for agricultural modernization [49]. This could also result in communal conflicts, as well as rural–urban migration [47]. The ability to control and manage agricultural land can be regarded as the future of food production, help to overcome the existing challenges within the agriculture sectors, and help accelerate sustainable development [50].

Another challenge facing global food system has to do with food wastage. In as much as there is a high demand for food, a substantial amount of food still go waste. It is estimated that about 30% of global food supplies are wasted, which represents about 1.2 billion tons of food [51, 52]. Even though the exact quantity of food wasted is still not clearly established due lack of a unified measurement system [53], there are still concerns about the possibility of food waste might threaten food security. Studies have shown if developed countries reduced post-harvest food waste by 50%, the number of undernourished people in developing countries can be reduced by up to 63 million [54, 55]. This then suggests that the reduction of food waste could also help manage global food security. Food utilization could be another component in the management of food security [56]. The proper utilization of food could help address issues of food allocation, as well as the dietary quality of food [57]. Given the multidisciplinary nature of food security, many disciplines have engaged in studies of food security issues and how these disciplines and sectors influence food security, including agriculture, anthropology, economics, nutrition, public policy, and sociology, as have numerous national and international governmental and nongovernmental agencies [58–61].

Another factor that has an impact on food security and poverty reduction is the movement of people across borders and even across continents. Migration has been considered one of the strategies for improving the standard of living [62]. Migration has the potential to increase a house's income through remittance and can also act as a cushioning mechanism at times of economic shocks [63]. Studies on migration have shown that about 300 million people are living outside their countries of origin in the last 40 years [64, 65]. Generally, hardships or

poverty are there driving forces that cause people to migrate from their countries of origin with the aim of finding a better financial opportunity [66]. There have been studies on the relationship between food security and migration, the association between migration and food security is mostly associated with the remittance, it is believed that remittance is an integral factor for household welfare in overcoming liquidity and unbinding insurance [67].

Migration can positively influence household expenditure through the remittance of basic needs like daily food. People also get a better sense of awareness of nutrition and dietary diversity through migration [66]. Food security conditions of households can also be improved through the migration of other family members by reducing the number of people that need to be fed within the household. However, Regmi & Paudel [68] argued that migration can endanger a household's food security status should the household lose a labor force through migration, this loss of family labor may not adequately be replaced by remittances. Few studies have also examined the linkage between migration and food security. The study by Atuoye et al [69] on the relationship between residential remittances and food security in Ghana from 1438 households revealed that migration and its associated remittance have a net positive impact on food security. However, Choithani [70] examined how internal migration influences food security among rural households in India, the study claimed that remittance from migrated family members has contributed positively to household food security by improving its purchasing power.

Environmental sustainability and government institutions play crucial roles in achieving Sustainable Development Goal Two (SDG 2), which aims to end hunger, achieve food security, improve nutrition, and promote sustainable agriculture. Environmental sustainability involves the protection and sustainable management of ecosystems, including forests, wetlands, and agricultural lands [71]. The preservation of ecosystems such as encouraging sustainable water use, promoting water-saving irrigation techniques, and safeguarding water quality support food production and also ensures the availability of natural resources necessary for agricultural production [72]. Environmental sustainability also helps in addressing climate change which can be vital for agriculture as it affects crop yields, water availability, and food production. Government institutions play a key role in developing and implementing policies that support sustainable agriculture and food security. This may includes integrating environmental sustainability into agricultural policies, creating frameworks for land-use planning, establishing regulations on agrochemical use, and promoting sustainable farming practices at national level. Governments can also affect food security through financial policies, providing incentives to farmers, adopting sustainable farming techniques, such as organic farming or agroforestry has huge role in the achievement of SDG 2 [73]. Governments can affect sustainable agriculture production by facilitating capacity-building programs and knowledge sharing platforms for farmers, such as training on sustainable agricultural practices, providing access to information and technology, and promoting farmer-to-farmer knowledge exchange [74]. Collaboration among governments, stakeholders, and communities is essential for effective implementation and long-term success. By integrating environmental sustainability into agricultural practices and leveraging the roles of government institutions, progress towards SDG 2 can be achieved. Prioritizing environmental sustainability and strengthening government institutions can also help to effectively achieve SDG 2. This requires a multi-dimensional approach involving policy coherence, capacity-building, stakeholder engagement, and sustainable farming practices at all levels of the agricultural system.

## The Sustainable Development Goal of zero hunger

The Global Hunger Index (GHI) Score reveals that many countries in Africa have a substantial percentage of their population living in extreme poverty [75]. The main causes of hunger in the African sub-region are mostly poverty, unemployment, conflicts, poor climatic conditions,

wars, insurgencies, corruption, and pre-harvest and post-harvest losses resulting from disease and pests. Conflicts and insurgencies play a devastating role in countries like Central African Republic (CAR), Somalia, Chad, and Southern Sudan. These countries have recorded high undernourishment rates [76–78]. The majority of African countries with high GHI also have a correspondingly high percentage of their population living in extreme poverty [75]. This still leaves millions of people eating less than they should with three-quarters of a billion people going a whole day without eating, thus falling far short of the Sustainable Development Goal Two (SDG 2) of ending hunger.

The world has made significant progress in addressing food insecurity in recent decades, with the prevalence of hunger decreasing from 15.7% in 1990–92 to 10.9% in 2014–16 [76, 79]. However, there is still much work to be done to achieve global food security. Recent estimates suggest that 795 million people suffer from chronic undernourishment and lack of access to safe and nutritious food [80]. This is particularly evident in developing countries, where the majority of people are in food-insecure households. Beyond the call to action in addressing world hunger, the Hot Springs conference was notable for its strong emphasis on nutrition and related food security issues [12]. The development of Sustainable Development Goal Two (SDG2) which aim at ending hunger by 2030 offers hope that the original 1943 Hot Spring conference vision of integrating agriculture and nutrition is finally receiving the required attention. SDG2 also offers a good foundation for the acceleration of progress in the reduction of malnutrition in so many dimensions [81]. Overcoming the challenge of hunger and achieving SDG2 will mean that a great increase in food production with adequate supply of food will be needed.

Given the multi-disciplinary and multi-sectoral nature of food security, studies have utilized different parameters to develop proxies to measure food security such as; (I) Household Consumption and Expenditure Surveys (HCES) which estimates household-level food consumption patterns, poverty, economic status, and consumer price indices of the household [56]. This gives information on food expenditures based on the monetary value of the quantity of food acquired by the household. (II) Food Consumption Score (FCS) is a measure of the composite score with a primary focus on dietary diversity, consumption frequency, and the relative nutritional relevance of various food groups [13]. This is usually used by the World Food Program to access food security on a regional scale. Studies from different parts of the world including Africa indicated a positive relationship between food consumption scores and the daily per capita kilocalorie intake [82–84]. (III) Household Dietary Diversity Score (HDDS), is another metric with much momentum in recent times regarding food security measures. The importance of this metric has been emphasized by Kennedy [85] to be an essential measure of food security in various households or even individuals. Household access to varieties of food in qualitative terms can also be measured using the dietary diversity score [86]. Household Dietary Diversity score reflects both food access and availability in the sense that the ability of a household to consume varieties of food groups will depend on the household purchasing power [87]. Studies have proven that increase in dietary diversity can be related to household food security and socioeconomic status [86]. (IV) The Household Food Insecurity Access Scale (HFIAS) is also another metric that has been used in recent years as a proxy to measure food security, this was developed as part of the Food and Nutrition Technical Assistance (FANTA II) Project [88]. The Household Food Insecurity Access Scale has also been employed to assess how households experienced problems with food access in the past.

## State of food security in Africa

According to the United Nations Food and Agriculture Organization (FAO), about one-third of the population in sub-Saharan Africa is undernourished, and the situation is especially

severe in rural and conflict-affected areas [89, 90]. The study by Cooper et al. [89] found that there is substantial diversity in the distribution of global severe food insecurity across countries and subnational regions. The study also reveals that, in the year 2020, sub-Saharan Africa is the region with the highest rates of severity in food insecurity, with about 20% of people over the corresponding threshold in at least one subnational area in every country with the except of Equatorial Guinea, Gabon, and Djibouti. It is estimated that, over 60% of the people living with food insecurity are in Sub-Saharan Africa with hunger estimated to worsen in this region if measures are not taken to tackle the issue of food insecurity [13]. This obviously does not come as a surprise since most undeveloped nations are in this part of the world given that the major contributing factor to food insecurity is poverty. In Kenya alone, it is estimated that about 1 million people were severely food insecure in the last five years [23], while about 8 million people have food insecurity issues in Ethiopia [91]. The main challenge facing Africa is the limited availability of food, many African countries have inadequate food production, leading to an over-dependence on imported food, which is often expensive and might be of poor quality [92]. This has been exacerbated by climate change, which has led to extreme weather events such as droughts, floods, and desertification, resulting in reduced crop yields.

Also, one key element of Africa's food security is the unavailability of food from either domestic production or the inability to access the global market, accessibility of food by consumers, affordability, and safety [93, 94]. Moreover, population growth has put increased pressure on existing resources, leading to greater food insecurity. In terms of food accessibility, in some regions, especially in rural areas, access to food is severely limited due to poverty, poor infrastructure, and a lack of resources [95]. In other parts of Africa, access to food is more available but still limited due to high prices, poor quality, and other factors [96]. Mali, Ghana, and South Africa are the only countries considered to be relatively secure in terms of food accessibility as at 2017; with Egypt losing its stability status while Nigeria and Gabon recorded undernourishment exceeding 10% of the population [3]. This means that the majority of African countries are dependent on food imports to fulfil their food availability needs, which implies food accessibility will be affected by their import capacity.

One of the challenges facing sub–Saharan African food security will have to do with low agriculture production. Agriculture activities in Africa have been characterized by low productivity, this is mainly attributed to climate change and variability [97], the lack of sustainable land and agronomic practices [98], and low technology [99]. Crop yield has been steadily on a decline due to factors such as poor soil quality, inefficient nutrient utilization, and as well as water management, these factors hugely affect agriculture operations [2]. Crop production in many parts of Africa is predominantly open field and rainfed, making it highly vulnerable to changes in climate conditions. Poor soil quality is a major obstacle to improving crop yield in many African smallholders' fields. Even agriculture in the United State of America is facing constraints such as the availability of arable land and freshwater. Agriculture production in the sub-Saharan region has been hugely affected by the increase in the price and demand of fertilizer, most of the continent's agriculture importation comes from Russia and Belarus [100]. The existence of Russia Ukraine war will mean the global supply of fertilizer and other agricultural products needed for crop production will be limited, resulting in lower agricultural productivity for fertilizer importers. Increasing food production among African countries with low crop yields will require an increase in agricultural investments (in machinery, irrigation, fertilizers, and pesticides) to modernize agriculture activities. These investments may have seen an upgrade to include the acquisition of large-scale land in the form of direct land leases or investments in land ownership [3].

At the same time, there are some encouraging signs. In recent years, African governments have taken steps to improve food security, such as investing in agricultural research and

development (planting for food for jobs flagship program in Ghana), implementing social protection programs, and investing in infrastructure [101]. African governments' commitments to regional and global treaties such as the Paris Agreement, Agenda 2030, and 2063 also present an opportunity to invest and adopt sustainable agriculture production technologies. The FAO has also reported that the number of people in Africa who are undernourished has been decreasing since 2000 [102]. Despite these gains, there is still much work to be done. Food insecurity remains a major challenge in Africa, and it is estimated that nearly 250 million people are still at risk of hunger and malnutrition [103]. In addition, climate change is expected to worsen the situation, with increased droughts, floods, and other extreme weather events projected to increase food insecurity in the region [104, 105]. In order to ensure food security for all in Africa, governments and international organizations must continue to prioritize investments in agricultural development, social protection, and infrastructure.

## Methodology

To achieve the purpose of the study, a Generalized Method of Moments was deployed as used by Gyimah & Yao [106] and Two-Staged Least Squares as used by Nguyen & Nasir [107]. The study uses GMM because it is able to deal with endogeneity issues and able to control and check any form of unknown arbitrary heteroscedasticity. The Two-Stage Least Squares serve as the robustness check. Therefore, the uniqueness of this study is in the relevance of the results. The study examines how food security, carbon emissions, and governance institutions affect each other in West Africa.

Eq 1 examines the effect of carbon emissions, governance institutions, and the control variables on food security. Food security has been one of the major battles of many developing countries. Government efforts are required massively to promote this agenda. However, the fight cannot be completed when the environment is in danger. In this respect, the first equation examines if whether carbon emissions and governance institutions promote food security or not.

$$lfs_{it} = \alpha_0 + \alpha_1 \lg_{it} + \alpha_2 lco_{2it} + \alpha_3 le_{it} + \alpha_4 lp_{it} + \alpha_5 lr_{it} + \alpha_6 lu_{it} + \varepsilon_{it} \tag{1}$$

Eq 2 explains the influence of food security and governance institutions on carbon emissions. Aside from the fight to promote food security, environmental sustainability is required to protect human continuity. Global warming has been one of the major concerns of states and non-state actors. Therefore, the result will show how governance institutions and food security are helping to solve the problem.

$$lco_{2it} = \alpha_0 + \alpha_1 \lg_{it} + \alpha_2 lfs_{it} + \alpha_3 le_{it} + \alpha_4 lp_{it} + \alpha_5 lr_{it} + \alpha_6 lu_{it} + \varepsilon_{it} \tag{2}$$

Eq 3 emphasizes on the effect of food security and carbon emissions on governance institutions. Government institutions in some West African countries have experienced political instability, economic instability, and institutional failure for years now. Many interventions were sought but the problems still exist now. The study focuses on how the independent variables and the control variables encourage the quality of governance in West Africa.

$$\lg_{it} = \alpha_0 + \alpha_1 lfs_{it} + \alpha_2 lco_{2it} + \alpha_3 le_{it} + \alpha_4 lp_{it} + \alpha_5 lr_{it} + \alpha_6 lu_{it} + \varepsilon_{it} \tag{3}$$

Eqs 4, 5 and 6 show how the indicators of food security are influenced by the independent variables and the control variables. The indicators are; agriculture, forestry and fishing, forest area, and food production index. The effect on these individual variables would help to know

how each variable is affected when standing alone.

$$lag_{it} = \alpha_0 + \alpha_1 \lg_{it} + \alpha_2 lco_{2it} + \alpha_3 le_{it} + \alpha_4 lp_{it} + \alpha_5 lr_{it} + \alpha_6 lu_{it} + \varepsilon_{it} \tag{4}$$

$$lfe_{it} = \alpha_0 + \alpha_1 \lg_{it} + \alpha_2 lco_{2it} + \alpha_3 le_{it} + \alpha_4 lp_{it} + \alpha_5 lr_{it} + \alpha_6 lu_{it} + \varepsilon_{it} \tag{5}$$

$$lfpi_{it} = \alpha_0 + \alpha_1 \lg_{it} + \alpha_2 lco_{2it} + \alpha_3 le_{it} + \alpha_4 lp_{it} + \alpha_5 lr_{it} + \alpha_6 lu_{it} + \varepsilon_{it} \tag{6}$$

## Data

The data for the study covers 16 West African countries. The data was obtained from World Development Indicators (https://databank.worldbank.org/source/world-development-indicators), spanning from 1990 to 2021. The variables are; food security (agriculture, forestry and fishing, forest area, food production index), carbon emissions, governance institutions (control of corruption, government effectiveness, political stability, regulation quality, rule of law, and voice and accountability), GDP, renewable energy, population growth, and urbanization. Table 1 gives the source of the variables, Table 2 explains the statistical description of the variables in their logged form, and Table 3 shows the correlational relationship between the variables.

## Results and discussion

### Panel unit root test, cross-section dependency test, and Panel cointegration Test

Panel unit root test is employed to assess if the variables are stationary for the study. The results present in Table 4 is the outcome for both the panel root test and the cross-section

**Table 1. Variables for the study.**

| Variables | Measurement | Source | Year |
|---|---|---|---|
| Agriculture, forestry and fishing (*lag*) | Percentage of GDP of agriculture, forestry and fishing | WDI | 1990–2021 |
| Forest area (*lfa*) | Percentage of land area | WDI | 1990–2021 |
| Food production index (*lfpi*) | Total edible and nutrients food crops | WDI | 1990–2021 |
| Governance institutions (lg) | Economic, political, and institutional governance | WDI | 1990–2021 |
| Carbon emissions ($lco_2$) | Metric tons per capita | WDI | 1990–2021 |
| Food security (*lfs*) | Food production index, agriculture, forestry, fishing, and forest area | WDI | 1990–2021 |
| Renewable energy consumption (*lr*) | Total percentage of renewable energy consumption | WDI | 1990–2021 |
| Population growth (*lp*) | Annual percentage of population growth | WDI | 1990–2021 |
| Urbanization (*lu*) | Percentage of total population | WDI | 1990–2021 |
| Economic growth (*le*) | GDP per capita | WDI | 1990–2021 |

**Table 2. Statistical description.**

| | Mean | Median | Max | Mini | Std. Dev |
|---|---|---|---|---|---|
| *lfs* | -1.014177 | -0.820688 | 0.936571 | -6.454644 | 1.166013 |
| $lco_2$ | -1.237679 | -1.241827 | -0.036488 | -6.311119 | 0.477702 |
| lg | -0.018625 | 0.129877 | 1.481775 | -4.924783 | 1.109729 |
| *le* | 22.71131 | 22.57956 | 26.93787 | 20.08839 | 1.497968 |
| *lr* | 4.142529 | 4.317421 | 4.523489 | 3.033991 | 0.373649 |
| *lp* | 0.944917 | 0.979423 | 1.532488 | -1.599003 | 0.279974 |
| *lu* | 1.346074 | 1.365722 | 1.905826 | -0.115404 | 0.255546 |

**Table 3. Correlation.**

|  | *lfs* | lg | *lco$_2$* | *le* | *lr* | *lp* | lug |
|---|---|---|---|---|---|---|---|
| *lfs* | 1 |  |  |  |  |  |  |
| lg | 0.166 | 1 |  |  |  |  |  |
| *lco$_2$* | -0.032 | 0.245 | 1 |  |  |  |  |
| *le* | -0.102 | -0.238 | 0.024 | 1 |  |  |  |
| *lr* | -0.442 | -0.740 | -0.214 | 0.484 | 1 |  |  |
| *lp* | -0.417 | -0.650 | 0.125 | 0.452 | 0.895 | 1 |  |
| *lu* | -0.409 | -0.555 | 0.003 | 0.299 | 0.868 | 0.914 | 1 |

dependency test. The outcomes show that the variables are stationary at first difference and the cross-section dependency test is significant for all the variables. The result presented in Table 5 is the outcome from the Panel cointegration Test. The outcome reveals that there is existence of cointegration among the variables.

## The generalized methods of moments estimation

The results in Table 6 show three dimensions, Firstly, the effect of carbon emissions and government institutions on food security. Secondly, the effect of food security and government institutions on carbon emissions. Final dimension shows the effect of food security and carbon emissions on government institutions. The main purpose is to establish how they affect each other in the promotion of food security in West Africa. The first dimension of the result indicates that, carbon emissions have no effect on food security. The amount of carbon emissions released into the atmosphere does not influence the level of food security in West Africa. However, government institutions reduce food security in West Africa. The government institutions represented in this study are; economic governance, political governance, and institutional governance. These institutions are expected to provide and implement policies that will promote food sustainability however since West Africa over the past years has faced institutional changes especially political and economic governance, these institutions are not strong enough to promote food security in the region [108]. The income level and urbanization promote food security in West Africa. Population growth and renewable energy reduce food security in the region. The result indicates that, the various institutions responsible for food sustainability are required to take advantage of urbanization and the economy to protect and improve on food sustainability.

**Table 4. Panel unit root test and cross-section dependency test.**

|  | Levin, Lin & Chu | | Im, Pesaran and Shin | | ADF—Fisher Chi-square | | Cross section |
|---|---|---|---|---|---|---|---|
|  | levels | 1st difference | Level | 1st difference | levels | 1st difference | Breusch-Pagan LM |
| *lco$_2$* | -1.582 | -14.73*** | -0.940 | -13.01*** | 37.88 | 179.1*** | 605.3*** |
| *lfs* | -10.117*** | -15.89*** | -1.231 | -2.782*** | 49.55*** | 68.99*** | 181.3*** |
| lg | -2.090 | -9.108*** | -1.784 | -5.830*** | 28.85 | 68.37*** | 179.8*** |
| *lp* | 0.843 | -5.594*** | 4.949 | -3.277*** | 52.90*** | 76.74*** | 1002.2*** |
| *lr* | -0.287 | -15.83*** | 1.992 | -13.88*** | 18.65 | 207.5*** | 930.3*** |
| *le* | 0.486 | -14.57*** | 5.683 | -15.46*** | 13.10 | 235.4*** | 2310.4*** |
| *lu* | -1.797 | -4.647*** | 0.064 | -6.562*** | 36.84 | 103.0*** | 900.8*** |

**Table 5. Panel cointegration Test.**

|  | Statistic | Weighted Statistics |
|---|---|---|
| Panel v-Statistic | -650.6948 | -1.417572 |
| Panel rho-Statistic | 1.903185 | 1.908747 |
| Panel PP-Statistic | -2.121717*** | -2.141328*** |
| Panel ADF-Statistic | -1.299685* | -1.339470* |
| Group rho-Statistic | 2.590930 |  |
| Group PP-Statistic | -2.206463*** |  |
| Group ADF-Statistic | -1.314886* |  |

The second dimension of the results shows that, food security has no significant effect on carbon emissions. This result corresponds to the effect of carbon emissions on food security in the first dimension of the results. However, government institutions promote carbon emissions. Beside government institutions having negative effect on food security in the first dimension of the results, it contributes to environmental degradation in the second dimension of the results. Since environmental regulations in developing countries are weak, firms in developed countries turn to take advantage to export their high emission technologies to these countries. The weak system of these countries is not able to restrict the use of these technologies resulting in environmental degradation. Therefore, the inability of government institutions to enforce environmental regulations has resulted this firms contributing to carbon emissions in West Africa. Income and population growth increase carbon emissions in West Africa. The population in the region keeps growing and the activities the growing population involves in affect the environment. Nevertheless, renewable energy promotes environmental quality. It is argued that, renewable energy is the best alternative to replace fossil fuel as a measure of protecting the environment [109] and the result of this study supports that argument.

The third dimension of the results further shows that, food security has negative and significant effect on government institutions. In the first dimension of the results, it could be seen

**Table 6. The effect of food security, carbon emissions, and governance institution.**

|  | $lfs$ | $lco_2$ | $lg$ |
|---|---|---|---|
| $lfs$ |  | -0.015282 | -0.241066*** |
|  |  | (-0.574204) | (-2.596515) |
| $lco_2$ | -0.311198 |  | 0.802148* |
|  | (-0.574204) |  | (1.873302) |
| $lg$ | -0.369241*** | 0.060335* |  |
|  | (-2.596515) | (1.873302) |  |
| $le$ | 0.289198*** | 0.046699** | 0.335537*** |
|  | (3.391316) | (2.379921) | (5.367830) |
| $lr$ | -2.317116*** | -0.758240*** | -2.321778*** |
|  | (-2.903250) | (-4.634387) | (-3.724793) |
| $lp$ | -1.670073* | 0.984409*** | -3.901719*** |
|  | (-1.752075) | (5.455641) | (-6.177279) |
| $lu$ | 1.971898* | -0.068063 | 4.037417*** |
|  | (1.726349) | (-0.263402) | (4.998773) |

***, **, * represent 1%, 5%, and 10% respectively

that government institutions reduce food security and the result in the third dimension is also indicating the negative effect of food security on government institutions. Carbon emissions has positive and significant effect on government institutions and government institutions have the same effect on carbon emissions in the second dimension of the results. This indicates that they influence each other in the same direction. Income level and urbanization improve government institutions, while renewable energy and population growth have negative significant effect on government institutions.

Food security can significantly deteriorate the environment due to the continuous depletion of the earth's resources. This could cause food security to have a substantial contribution to greenhouse gas emissions [30]. Study by Ashraf and Javed [30] on the relationship between food security, institutional quality and environmental degradation revealed that food security increases environmental deterioration while institutional quality reduces environmental deterioration. The study further revealed that food security through human capital improves environmental sustainability. In addition, Study by Oyelami et al [110] revealed that climate change plays a crucial role in sustainable and uncovered the negligible role of institutions in promoting food security. Study by Soko et al [111] further highlighted on the mediating role of institutional quality in impacting food security. The results from the ARDL estimation revealed that population growth reduces food security while granger causality revealed that there is unidirectional causal effect from food security to economic growth [112]. However, our findings revealed no effect of food security on carbon emissions and vice versa.

## The Two-Stage Least Squares estimation

In order to verify the findings of the study, Two-Stage Least Squares is employed as the robustness check and the result is represented in Table 7. The result of the robustness check verifies the results of the main model thus GMM. The result from the robustness check indicates negative feedback effect of governance institutions and food security. Food security and carbon emissions are not significant to explain each other however, governance institutions and carbon emissions have positive feedback effects. The signs of the effects of the GMM are the same

**Table 7. Two-Stage Least Square.**

|  | *lfs* | *lco₂* | *lg* |
|---|---|---|---|
| *lfs* |  | -0.015282 | -0.241066*** |
|  |  | (-0.574204) | (-2.596515) |
| *lco₂* | -0.311198 |  | 0.802148* |
|  | (-0.574204) |  | (1.873302) |
| *lg* | -0.369241*** | 0.060335* |  |
|  | (-2.596515) | (1.873302) |  |
| *le* | 0.289198*** | 0.046699** | 0.335537*** |
|  | (3.391316) | (2.379921) | (5.367830) |
| *lr* | -2.317116*** | -0.758240*** | -2.321778*** |
|  | (-2.903250) | (-4.634387) | (-3.724793) |
| *lp* | -1.670073* | 0.984409*** | -3.901719*** |
|  | (-1.752075) | (5.455641) | (-6.177279) |
| *lu* | 1.971898* | -0.068063 | 4.037417*** |
|  | (1.726349) | (-0.263402) | (4.998773) |

***, **, * represent 1%, 5%, and 10% respectively

**Table 8. Effect of carbon emissions and governance institutions on food security indicators.**

|  | *lfpi* | *lfa* | *lag* |
|---|---|---|---|
| $lco_2$ | 0.458417*** | 1.705154*** | 0.205761*** |
|  | (3.867262) | (4.727998) | (2.469868) |
| lg | -0.155845*** | 0.283121*** | 0.004134 |
|  | (-4.825390) | (2.881258) | (0.182140) |
| le | 0.196331*** | -0.064646 | -0.027446* |
|  | (9.440302) | (-1.021661) | (-1.877787) |
| lr | 0.240309 | 1.612936*** | 0.916645*** |
|  | (1.261990) | (2.784036) | (6.849455) |
| lp | -1.511675*** | -1.249092* | 0.464207*** |
|  | (-6.834522) | (-1.856159) | (2.986273) |
| lu | 0.681971*** | 0.867572* | -0.118014 |
|  | (4.062768) | (1.698762) | (-1.000364) |

***, **, * represent 1%, 5%, and 10% respectively

as the signs of the robustness. However, there is a slight change in the magnitude. Nevertheless, the change in the magnitude does not affect the effects on the variables.

## The long-term estimation of the food security indicators

The effects represented in Table 8 show how carbon emissions, governance institutions, and the control variables influence the indicators used to represent food security. The effects indicate that carbon emissions positively affect food production index, agriculture, forestry and fishing, and forest area. Our finding is supported by Raihan et al [113] which study on Bangladesh revealed that carbon emissions causes forest area. However, governance institution has mixed effects, thus it has negative effect on food production index and positive effect on forest area but insignificant effect on agriculture, forestry and fishing. Renewable energy has positive effect on forest area and agriculture, forestry and fishing but is not significant to explain food production index. Income has positive effect on food production index and negative effect on agriculture, forestry and fishing but insignificant effect on forest area. Population growth has negative effect on food production index and forest area but positive effect on agriculture, forestry and fishing. Urbanization has positive effect on food production index and forest area but insignificant to explain agriculture, forestry and fishing.

Several studies have explored the effects of these variables and the literature has captured various effects. According to the study by Raihan et al [113] forest area helps to promote environmental sustainability by reducing carbon emissions. Study by Raihan [114] indicated that 1% rise in forest area and agriculture productivity would respectively reduce carbon emissions by 3.46% and 0.20%. Another study in Malaysia also indicated that increase in forest area reduces carbon emissions [115]. However, The findings of Ashraf and Javed [30] on food production index and environmental deterioration revealed that the increase in food production index may cause more environmental deterioration. Study by Wang et al [116] revealed that increase in agricultural specialization has positive effects on agricultural carbon emissions as a result of rise in agricultural external inputs. Raihan [114] also revealed that forest area granger-causes economic growth and renewable energy.

## Conclusion and policy implications

Increasing staple food production on the back of the current global crisis is necessary for attaining a sustainable development and national food security. Food security is an important strategy for reduction of hunger, improving good health, and promoting economic growth. Achieving global food security and environmental sustainability mean having a sustainable agriculture production and the delivery of economic welfare for society. In general, food security in Africa has been declining over the past few decades due to a variety of factors, such as population growth, bad governance, climate change, conflicts, and economic inequality. These factors and food security have a mediating effect on each other. The elimination of food insecurity, poverty as well as the management of natural resources will require major policy implementation which intern will require a better assessment of the controlling factors mediating food security for sustainable development.

Two different models are used to examine the effect of food security, carbon emissions, and governance indicators on each other. Generalized Method of Moments is employed as the main model and Two-Stage Least Squares is employed as a robustness check. The purpose is to establish how food security can be promoted in a sustainable environment. The findings of the study are in three folds. The first fold indicates that carbon emissions has no effect on food security, government institutions reduce food security, income level and urbanization promote food security, and population growth and renewable energy reduce food security. The second fold shows that food security has no significant effect on carbon emissions, while government institutions promote carbon emissions. Income level and population growth have positive and significant effect on carbon emissions, while renewable energy promotes environmental sustainability. The third fold indicates that food security has negative and significant effect on government institutions, carbon emissions have positive and significant effect on government institutions, income level and urbanization improve government institutions, and renewable energy and population growth have negative and significant effect on government institutions.

Based on the result of the study, these policy implications are recommended in order to promote food security. The result of the study revealed that government institutions has negative effect on food security. To make recommendation regarding this result, there must be respect for rule and law. The government institutions must have the supremacy to implement rules and regulations that would promote food security. Government institutions must be independent to take relevant decisions on food security. In addition, the result further revealed that government institutions cause carbon emissions and responding to this issue, government institutions must be bold to make decisions and policies that would put restrictions on the use of high carbon emissions technologies. Environmental laws must be reviewed to meet the current environmental situation to promote environmental sustainability.

## Author Contributions

**Conceptualization:** Justice Gyimah.

**Data curation:** Justice Gyimah.

**Formal analysis:** Justice Gyimah, Louis K. M. Nibonmua.

**Investigation:** Justice Gyimah.

**Methodology:** Justice Gyimah, Benjamin M. Saalidong.

**Supervision:** Justice Gyimah.

**Visualization:** Justice Gyimah, Benjamin M. Saalidong.

**Writing – original draft:** Justice Gyimah.

**Writing – review & editing:** Justice Gyimah, Benjamin M. Saalidong, Louis K. M. Nibonmua.

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
