## [Decision Letter · Decision Letter 0]

2 Aug 2023

PONE-D-23-20127

The Battle to Achieve Sustainable Development Goal Two: the Role of Environmental Sustainability and Government Institutions

PLOS ONE

Dear Dr. GYIMAH,

Thank you for submitting your manuscript to PLOS ONE. After careful consideration, we feel that it has merit but does not fully meet PLOS ONE's publication criteria as it currently stands. Therefore, we invite you to submit a revised version of the manuscript that addresses the points raised during the review process.ACADEMIC EDITOR: Two expert reviewers have reviewed your manuscript and the comments/suggestions are attached to this email. Kindly attend to their comments/suggestions.

We look forward to receiving your revised manuscript.

Kind regards,

Olutosin Ademola Otekunrin

Academic Editor

PLOS ONE

Journal Requirements:

Monitoring and projecting global hunger: Are we on track? - https://doi.org/10.1016/j.gfs.2021.100568

In your revision ensure you cite all your sources (including your own works), and quote or rephrase any duplicated text outside the methods section. Further consideration is dependent on these concerns being addressed.

Reviewers' comments:

Reviewer's Responses to Questions

**Comments to the Author**

1. Is the manuscript technically sound, and do the data support the conclusions?

Reviewer #1: Partly

Reviewer #2: Yes

2. Has the statistical analysis been performed appropriately and rigorously? 

Reviewer #1: Yes

Reviewer #2: Yes

3. Have the authors made all data underlying the findings in their manuscript fully available?

Reviewer #1: No

Reviewer #2: Yes

4. Is the manuscript presented in an intelligible fashion and written in standard English?

Reviewer #1: Yes

Reviewer #2: Yes

5. Review Comments to the Author

Reviewer #1: Reviewer’s comments

The study aimed to examine the role of environmental sustainability and government institutions in sustainable development goal two in West Africa.

Introduction

Paragraph 2: The sentence “It was estimated that the number of food-insecure people could exceed 1 billion by the end of 2020 (World Health Organization, 2020). and This number was estimated to reach 132 million by the end of 2020 (World Health Organization, 2020).” needs to be updated with recent information.

The author should focus their discussion on achieving sustainable development goal 2, and not poverty.

The introduction is lacking an important aspect, “the need for the study”. It is critical to clearly state the contribution of the study to the body of knowledge by clearly showing the research gap.

2 Literature Review

The author should correct the citation patterns in this heading. For instance, Similarly, the work of (Amwata et al., 2016) reveals that, in marginal areas of Kenya, households with agropastoral livelihood are less vulnerable in terms of food security as compared to their pure pastoral counterparts. (Turyahabwe et al., 2013) studies on how wetland resources contribute to household food security in agroecological zone in Uganda. There are many of these in the text.

The literature review focused on previous work without connection with the current study. it is expected that the authors will have a paragraph at the end to bring out the need for the current study to show the gaps in the literature.

The author should give the concept and theories in support of the work.

2.1 The SDG of zero hunger

Define SDG in the first use. Same as DG.

Correct the in-text citation. For instance, (G. L. Kennedy, 2009).

2.2 State of Food Security in Africa

The author should correct the citation patterns in this heading. For instance, the study by (Cooper et al., 2021) found that there is substantial diversity in the distribution of global severe food insecurity across countries and subnational regions.

Materials

The author should correct the citation patterns in this heading. For instance, to achieve the purpose of the study, a Generalized Method of Moments was deployed as used by (Gyimah & Yao, 2022) and Two-Staged Least Squares as used by (Nguyen & Nasir, 2021).

Data

It was stated in the abstract that the data covers 1996 to 2021, while it was stated in the methodology that it covers 1990 to 2021. The authors need to correct this.

How the data were arranged and used needs to be explained. For instance, how did the author merge the data from the 19 countries before the analysis?

The data used need to be described in terms of how WDI collected the data and so on.

Table 2 Statistical Description

The results here show that the data are logged. Clearly state if the data are logged or not.

Table 3 Correlation

There are some variables with high correlation. How did the authors take care of these variables?

Results and Discussions

The results of the findings were not well discussed. The author should explain each of the significant variables, state the reasons behind the results and relate them to the literature. For instance, how did carbon emissions positively affect the food production index, agriculture, forestry and fishing, and forest area? This and other results need to be critically explained for readers to understand.

Considering that the countries in West Africa have different policies, Governance institutions, Renewable energy consumption and population growth, among others. how did the authors take care of these differences? In addition, a specific investigation and analysis for each country selected would be better. This will enable us to understand what affects food insecurity in each nation.

Conclusion and policy implication

The conclusion should be concise; it is not a summary of the work. Policy implications should be directly from the findings of the study.

Reviewer #2: Summary

Food security remains one of the top priorities of the United Nations Sustainable Development Goals (SDG), which aims to alleviate hunger (zero Hunger) by 2030. Eliminating food insecurity, hunger and malnutrition by 2025 has been the main target of the African Union's Agenda 2063. Achieving this goal comes with its own challenges considering the need to feed more people from the expected increasing global population coupled with the impacts of climatic changes on food production. Population growth has brought a high demand for limited world resources, most notably food supplies. In the era of globalization coupled with the projected increase in world population, the need to attain global food security and poverty reduction becomes a global challenge for humanity in achieving sustainability. Population density and climate change threaten the second sustainable development goal (zero hunger); attaining food and its related environmental security comes with significant challenges. The author, therefore, explores the current state of food security within sub-Saharan Africa and the connection of food security with various contributing factors such as economic growth, governance, geopolitical, climatic, population, and environmental factors and the effect of environmental sustainability and government institutions on food security in West Africa with data from 1996 to 2021, two models are employed using the Generalized Method of Moments as the primary model and Two-Stage Least Squares for the robustness check. Findings from this study revealed that carbon emissions representing environmental sustainability have no direct significant effect on food security, while government institutions reduce food security. Also, income and urbanization promote food security, while renewable energy and population growth reduce food security.

Major comments

The methods, results, and discussion are well documented with good clarity and are scientific.

Minor comments

1. In the abstract, the authors should do some grammar checks on some tenses (appropriate use of past tense).

2. The references should be improved upon and should be consistent across the manuscript. In some cases Reference were omitted. Intext references, where the author’s initial appears should be corrected appropriately. e.g undernourished (P. Sharma et al., 2016).

Also, sentences like “the study by (Frelat et al., 2016) reveals that over 30% of rural households across 17 countries within the sub–Saharan Africa are still struggling with food insecurity” where brackets enclose author’s name within a statement, such brackets should enclose only the year e.g Frelat et al. (2016). This correction should be effected across the whole manuscript.

In their study, it was observed that households go through a transitional phase of inadequate food security during planting and growing seasons to fully secured food security during periods of harvesting.

This statement “The increase in population growth certainly has a connection to food security, ranging from the changes in human diets to the methods and ways through which food is produced. For instance, the increase in population will also mean more people are becoming wealthier and thus will be able to eat more and regularly or more people turn to opt for more resource-intensive food, thereby leading to the decline of resources needed for food production” has no reference. (page 13, paragraph 2)

This statement “The proper utilization of food could help address issues of food allocation, as well as the dietary quality of food. Given the multidisciplinary nature of food security, many disciplines have engaged in studies of food security issues and how these disciplines and sectors influence food security, including agriculture, anthropology, economics, nutrition, public policy, and sociology, as have numerous national and international governmental and non-governmental agencies”. Has no reference (page 15, paragraph 1).

Last paragraph on page 17 has no reference- “Also, one key element of Africa's food security is the unavailability of food from either domestic production or the inability to access the global market, accessibility of food by consumers, affordability, and safety”

The last paragraph on page 19, in the literature review section, just before the methodology has no reference, here “Food insecurity remains a major challenge in Africa, and it is estimated that nearly 250 million people are still at risk of hunger and malnutrition. In addition, climate change is expected to worsen the situation, with increased droughts, floods, and other extreme weather events projected to increase food insecurity in the region”.

Paragraph 1 of the result and discussion section on page 22 has no reference “These institutions are expected to provide and implement policies that will promote food sustainability however since West Africa over the past years has faced institutional changes especially political and economic governance, these institutions are not strong enough to promote food security in the region”.

Similarly, most information in the result and discussion section was not properly referenced.

“Carbon emissions has positive and significant effect on government institutions and government institutions have the same effect on carbon emissions in the second dimension of the results. This indicates that they influence each other in the same direction. Income level and urbanization improve government institutions, while renewable energy and population growth have negative significant effect on government institutions”. Can this statement be compared with previous studies?

Conclusion/recommendation

Generally, the study titled was well-researched, very relevant, scientific, and very informative, it raised pertinent issues regarding the impact of government institutions and environmental sustainability on achieving the “zero hunger” which is the Sustainable Development Goal 2.

The authors demonstrated excellent mastery of the study. However, few minor corrections are required as highlighted.

6. PLOS authors have the option to publish the peer review history of their article (what does this mean?). If published, this will include your full peer review and any attached files.

Reviewer #1: **Yes: **Ridwan Mukaila

Reviewer #2: No

---

## [Author Response · Author response to Decision Letter 0]

8 Aug 2023

Response to Reviewers’ Comments 

Reviewer #1

Comment: Paragraph 2: The sentence “It was estimated that the number of food-insecure people could exceed 1 billion by the end of 2020 (World Health Organization, 2020). and This number was estimated to reach 132 million by the end of 2020 (World Health Organization, 2020).” needs to be updated with recent information. 

Response: Thank you for the comment. The paragraph has been revised and updated

Comment: The author should focus their discussion on achieving sustainable development goal 2, and not poverty. 

Response: Thank you for pointing this out. The manuscript has been revised accordingly 

Comment: The introduction is lacking an important aspect, “the need for the study”. It is critical to clearly state the contribution of the study to the body of knowledge by clearly showing the research gap. 

Response: Thank you for the comment. The introduction has been revised accordingly

Comment: The author should correct the citation patterns in this heading.

Response: Thank you for the comment. The citations in the manuscript has been corrected accordingly

Comment: The literature review focused on previous work without connection with the current study. it is expected that the authors will have a paragraph at the end to bring out the need for the current study to show the gaps in the literature. 

Response: Thank you for the comment. The literature review has been revised and updated to reflect the objective and aim of the current study

Comment: The author should give the concept and theories in support of the work

Response: Thank you for the comment. The concept and theories of the study has been provided to support the study

Comment: It was stated in the abstract that the data covers 1996 to 2021, while it was stated in the methodology that it covers 1990 to 2021. The authors need to correct this. 

Response: Thank you for pointing this out. The authors have corrected mistake. 

Comment: How the data were arranged and used needs to be explained. For instance, how did the author merge the data from the 19 countries before the analysis? 

Response: Thank you for the comment. The data used for the study is panel data and the authors used Excel to merge the data.

Comment: The data used need to be described in terms of how WDI collected the data and so on.

Response: Thank you for the comment. The authors have described the measurement of the variables and incorporated them in Table 1.

Comment: The results here show that the data are logged. Clearly state if the data are logged or not. 

Response: Thank you for the comment. The authors have stated the form of the statistical descriptive data in the data section (logged form)

Comment: There are some variables with high correlation. How did the authors take care of these variables? 

Response: Thank you for the comment. The authors did preliminary tests (cross-section dependency test, panel unit root test, and panel cointegration test) to check the suitability of the variables and the results revealed that the variables are suitable for the study.

Comment: The results of the findings were not well discussed. The author should explain each of the significant variables, state the reasons behind the results and relate them to the literature. For instance, how did carbon emissions positively affect the food production index, agriculture, forestry and fishing, and forest area? This and other results need to be critically explained for readers to understand. 

Response: Thank you for pointing this out. The authors have explained and backed their findings with literature as suggested

Comment: Considering that the countries in West Africa have different policies, Governance institutions, Renewable energy consumption and population growth, among others. how did the authors take care of these differences? In addition, a specific investigation and analysis for each country selected would be better. This will enable us to understand what affects food insecurity in each nation. 

Response: Thank you for the comment. The authors did preliminary tests to examine if the variables are well presented for the study. The tests show that the variables are well represented and suitable for panel study. In addition, the study did not include country base result because of the objective of the study thus to assess regional effect.

Comment: The conclusion should be concise; it is not a summary of the work. Policy implications should be directly from the findings of the study. 

Response: Thank you for the comment. The conclusion and policy implications has been revised accordingly to reflect the findings of the study

Reviewer #2

Comment: In the abstract, the authors should do some grammar checks on some tenses (appropriate use of past tense).

Response: Thank you for the comment. Grammar checks and all necessary corrections has been carried out.

Comment: The references should be improved upon and should be consistent across the manuscript

Response: Thank you for the comment. Citations in the entire manuscript has been corrected accordingly and necessary reference has been provided to support the claims made.

Comment: Are there other findings that support or contradict this assertion?

Response: Thank you for the comment. The section has been revised. 

Comment: Any contrary/similar opinions from previous findings with references?

Response: Thank you for the comment. The authors have addressed this issue and relevant findings have been cited.

---

## [Decision Letter · Decision Letter 1]

29 Aug 2023

The Battle to Achieve Sustainable Development Goal Two: the Role of Environmental Sustainability and Government InstitutionsPONE-D-23-20127R1Dear Dr. Gyimah,We re pleased to inform you that your manuscript has been judged scientifically suitable for publication and will be formally accepted for publication once it meets all outstanding technical requirements. Within one week, you'll receive an e-mail detailing the required amendments. When these have been addressed, you'll receive a formal acceptance letter and your manuscript will be scheduled for publication. An invoice for payment will follow shortly after the formal acceptance. To ensure an efficient process, please log into Editorial Manager at http://www.editorialmanager.com/pone/, click the Update My Information link at the top of the page, and double check that your user information is up-to-date. If you have any billing related questions, please contact our Author Billing department directly at author billing@plos.org. If your institution or institutions have a press office, please notify them about your upcoming paper to help maximize its impact. If they'll be preparing press materials, please inform our press team as soon as possible -- no later than 48 hours after receiving the formal acceptance. Your manuscript will remain under strict press embargo until 2 pm Eastern Time on the date of publication. For more information, please contact onepress@plos.org.

Kind regards,

Olutosin Ademola Otekunrin

Academic Editor

PLOS ONE

Additional Editor Comments (optional):

Reviewers' comments:

Reviewer's Responses to Questions

**Comments to the Author**

1. If the authors have adequately addressed your comments raised in a previous round of review and you feel that this manuscript is now acceptable for publication, you may indicate that here to bypass the “Comments to the Author” section, enter your conflict of interest statement in the “Confidential to Editor” section, and submit your "Accept" recommendation.

Reviewer #1: All comments have been addressed

Reviewer #2: All comments have been addressed

2. Is the manuscript technically sound, and do the data support the conclusions?

Reviewer #1: Yes

Reviewer #2: Yes

3. Has the statistical analysis been performed appropriately and rigorously? 

Reviewer #1: Yes

Reviewer #2: Yes

4. Have the authors made all data underlying the findings in their manuscript fully available?

Reviewer #1: Yes

Reviewer #2: Yes

5. Is the manuscript presented in an intelligible fashion and written in standard English?

Reviewer #1: Yes

Reviewer #2: Yes

6. Review Comments to the Author

Reviewer #1: (No Response)

Reviewer #2: all areas of concern raised has been addressed by the author. The manuscript may proceed for publication based on the judgement of the chief-editor.

7. PLOS authors have the option to publish the peer review history of their article (what does this mean?). If published, this will include your full peer review and any attached files.

Reviewer #1: **Yes: **Ridwan Mukaila

Reviewer #2: No

---

## [Editor Report · Acceptance letter]

6 Sep 2023

PONE-D-23-20127R1 

The Battle to Achieve Sustainable Development Goal Two: the Role of Environmental Sustainability and Government Institutions. 

Dear Dr. Gyimah:

I'm pleased to inform you that your manuscript has been deemed suitable for publication in PLOS ONE. Congratulations! Your manuscript is now with our production department. 

Kind regards, 

on behalf of

Dr. Olutosin Ademola Otekunrin 

Academic Editor

PLOS ONE